# Influence of Preparation Design, Marginal Gingiva Location, and Tooth Morphology on the Accuracy of Digital Impressions for Full-Crown Restorations: An In Vitro Investigation

**DOI:** 10.3390/jcm9123984

**Published:** 2020-12-09

**Authors:** Selina A. Bernauer, Johannes Müller, Nicola U. Zitzmann, Tim Joda

**Affiliations:** 1Department of Reconstructive Dentistry, UZB University Center for Dental Medicine Basel, University of Basel, 4058 Basel, Switzerland; selina.bernauer@unibas.ch (S.A.B.); n.zitzmann@unibas.ch (N.U.Z.); 2Private Practice, 80634 Munich, Germany; dr.johannes.a.mueller@gmail.com

**Keywords:** fixed prosthodontics, full crown, tooth preparation, intraoral optical scanning (IOS), digital dentistry

## Abstract

(1) Background: Intraoral optical scanning (IOS) has gained increased importance in prosthodontics. The aim of this in vitro study was to analyze the IOS accuracy for treatment with full crowns, considering possible influencing factors. (2) Methods: Two tooth morphologies, each with four different finish-line designs for tooth preparation and epi- or supragingival locations, were digitally designed, 3D-printed, and post-processed for 16 sample abutment teeth. Specimens were digitized using a laboratory scanner to generate reference STLs (Standard Tessellation Language), and were secondary-scanned with two IOS systems five times each in a complete-arch model scenario (Trios 3 Pod, Primescan AC). For accuracy, a best-fit algorithm (Final Surface) was used to analyze deviations of the abutment teeth based on 160 IOS-STLs compared to the reference STLs (16 preparations × 2 IOS-systems × 5 scans per tooth). (3) Results: Analysis revealed homogenous findings with high accuracy for intra- and inter-group comparisons for both IOS systems, with mean values of 80% quantiles from 20 ± 2 μm to 50 ± 5 μm. Supragingival finishing lines demonstrated significantly higher accuracy than epigingival margins when comparing each preparation (*p <* 0.05), whereas tangential preparations exhibited similar results independent of the gingival location. Morphology of anterior versus posterior teeth showed slightly better results in favor of molars in combination with shoulder preparations only. (4) Conclusion: The clinical challenge for the treatment with full crowns following digital impressions is the location of the prospective restoration margin related to the distance to the gingiva. However, the overall accuracy for all abutment teeth was very high; thus, the factors tested are unlikely to have a strong clinical impact.

## 1. Introduction

Continuous technical development has expanded opportunities in reconstructive dentistry and prosthodontics [1]. In particular, intraoral optical scanning (IOS), computer-aided design, and computer-aided manufacturing (CAD/CAM) have fostered complete digital workflows for the treatment of fixed dental prostheses (FDPs) [2]. IOS has become indispensable in everyday dental practice, in university education [3,4], and in dental laboratories [5,6,7]. 

Digital impressions have been proven to be more time-efficient compared to conventional impressions, and a majority of patients have preferred the digital impression technique rather than the conventional approach with plastic materials [8,9,10]. At the same time, IOS has simplified the process chain between dentist and dental technician. Complete digital workflows have rendered various work steps superfluous such as tray preparation, disinfecting, shipping of the conventional impression, and further preparations for the fabrication of gypsum dental casts [11]. IOS technology offers new possibilities for clinical routine in selected indications, especially in the field of fixed prosthodontics. By taking a digital impression, the intraoral situation is visually recorded with neither the mucosa nor the teeth needing to be physically touched. This prevents possible gingival displacement or tooth movement from the application of conventional elastomeric impression material [12,13,14]. IOS is also advantageous for the treatment of periodontally compromised dentitions with recessions, enlarged interdental spaces, and dental undercuts (such as pontics or cantilevers), which make an accurate impression difficult [15]. Additionally, IOS opens the door to chairside CAD/CAM systems that could offer treatment protocols with single-unit restorations in one clinical session [16]. In contrast to conventional impressions, technical factors must be taken into account when using a digital approach. IOS requires a direct line of sight on the object in order to create 3D surface files, which are known as standard tessellation language (STL) [17]. Additional studies have shown that a supragingival preparation margin in the impression is more accurate [17,18].

Besides the technical development related to digital impressions, the finish-line design for tooth preparation has remained a crucial aspect for the abutment tooth [19,20]. Are the same finish-line designs applicable for full-crown restorations, or are adjustments required to facilitate the application of IOS? The challenge is now to analyze the existing parameters of tooth preparation in order to identify the best design for an accurate IOS and further STL processing for clinically acceptable restorations, while considering minimal invasiveness combined with modern materials and adhesive luting technology [21].

The aim of this in vitro study was to analyze the influence of different finish lines for complete crown preparations, their locations related to the gingival margin, and tooth morphology on the accuracy of digital impressions. The null hypotheses tested were that the IOS accuracy does not depend on the finish-line design (tangential, narrow chamfer, wide chamfer, and shoulder), the gingival positioning of the finishing line (epi- and supragingival), or on tooth morphology (incisor and molar); secondly, there is no difference in performance between the IOS systems used (Trios 3 Pod, 3Shape, Copenhagen, Denmark and Cerec Primescan AC, Dentsply Sirona, Bensheim, Germany).

## 2. Materials and Methods

A maxillary dental training model was used as reference (Dental Model AG-3, Frasaco, Tettnang, Germany). A maxillary central incisor (FDI 11) was selected to represent the anterior tooth morphology, while a first maxillary molar (FDI 16) was chosen to represent posterior sites. Based on a standardized complete crown preparation, two typodonts were manually prepared with a supragingival finishing line, 0.4 mm chamfer, and a 4–6° convergence angle. Substance removal was incisal 2.0 mm, palatal 1.0 mm, and labial 1.0–1.5 mm for tooth 11, and occlusal 1.5 mm and labial 1.0 mm, palatal 1.0 mm, and interdental 1.0 mm for the molar. All practical work steps were performed by the same operator (S.B.), a postgraduate prosthodontic resident, and each step was supervised by a senior clinician and board-certified prosthodontist (J.M.).

Both prepared typodonts were digitized with a laboratory desktop scanner (Series 7, Institute Straumann AG, Basel, Switzerland), and these served as the basis for the digital designs of the virtual modifications to create the test specimens, involving four different finish-line designs for both morphologies. These designs were digitally computed with the software Geomatic Design X (3D Systems, Rock Hill, SC, USA) and saved as STL files. The following finish-line designs were applied: tangential, narrow chamfer (0.4 mm), wide chamfer (0.8 mm), and shoulder (0.8 mm). Each design was applied in an epigingival or a 1.0 mm supragingival position, resulting in a total of eight tooth preparations for the anterior and another eight for the posterior region. Figure 1 displays the study setup with 16 different specimens (Figure 1).

Finally, the 16 virtual tooth preparations were 3D-printed for the production of standardized replicas (3D-Printer Objet260 Connex2, Stratasys, Eden Prairie, MN, USA). The color of the rubber-like material used was a mixture of Vero White Plus RGD 835 and Tango Black Plus FLX 980. This mixture resulted in the color DM8515 (Stratasys, Eden Prairie, MN, USA). All 3D-printed teeth were mounted in the reference model and manually finalized with diamond burs (Intensive SA, Montagnola, Switzerland) and Sof-Lex discs (3M ESPE AG, Saint Paul, MN, USA) to achieve an exact finishing line and a smooth surface. In order to avoid potential deviations due to printing errors and to visualize the manual corrections, all teeth were removed from the reference model and digitized with the same laboratory desktop scanner that was used for the initial digitalization.

Successively, all 16 3D-printed teeth were remounted in the reference model and scanned by one experienced operator (S.B.) with two IOS systems (Trios 3 Pod and Cerec Primescan AC). Each preparation, including adjacent teeth, was captured five times in order to minimize potential scanning errors. The scans were carried out according to the manufacturer’s recommendations.

For accuracy of analysis, a total of 160 IOS-STLs (16 specimens × 2 IOS-systems × 5 scans = 160 STLs) were then superimposed to the corresponding original reference STLs with the software Final Surface (GFaI e.V., Berlin, Germany). A best-fit algorithm was applied for deviation analysis in order to minimize the distances between the two surfaces being compared. Here, the distance to the surface of the IOS-STL to be examined was considered for all surfaces of the matching reference STL. Scanning data beyond 2 mm from the finishing lines were digitally cut to guarantee an accurate fine registration. Trimmed scan data obtained from five scans by each IOS were paired, and these pairs were inspected (STL-1 vs. STL-2, STL-1 vs. STL-3, STL-1 vs. STL-3, etc.). Deviations between polygons formed by the point cloud constituting the two superimposed scans were calculated, and the distance data of all superimposed pairs were summarized [22].

Numerical variables of interest were descriptively analyzed with sample means for 80% quantiles including standard deviation. Since the IOS-STLs under investigation appeared with a plus or minus of data points compared to the reference STLs, the use of 80% quantiles ensured error minimization regarding “too small” and “too large” areas for deviation analysis. Statistics were carried out using R 4.0.3 (The R Project for Statistical Computing, Vienna, Austria), and a significance level was set at 0.05.

## 3. Results

Trios 3 Pod and Primescan AC could successfully capture all selected preparations as tangential, narrow and wide chamfer, and shoulder, respectively (Figure 2). Intra- and intergroup analyses comparing both IOS systems revealed homogenous results with high accuracy representing mean values of 80% quantiles ranging from 20 ± 2 μm to 50 ± 5 μm throughout all tested abutment teeth (Table 1 and Table 2). Supragingival finishing lines demonstrated significantly higher accuracy than epigingival margins when comparing preparation designs against each other (*p <* 0.05), whereas tangential preparations exhibited similar results independent of the gingival location of the finishing line. Morphology of anterior versus posterior teeth showed slightly better results in favor of molars in combination with shoulder preparations only.

## 4. Discussion

The aim of this in vitro study was to analyze IOS accuracy for complete crown restorations, considering maxillary incisor and molar tooth morphologies with four different finish-line designs (tangential, narrow chamfer, wide chamfer, and shoulder) in epi- and supragingival margin positions. The results demonstrated that all specimens were successfully digitized with high accuracy independently of the IOS device used. However, the supragingival finishing lines were captured significantly better than the epigingivally located margins. Therefore, the hypothesis that IOS accuracy does not depend on any of the factors listed above was partially rejected.

IOS technology offers new possibilities for clinical routine in selected indications, especially in the field of fixed prosthodontics with all the advantages mentioned above. In the present study, the position of the finishing line with respect to the gingiva showed differences between epi- and supragingival margins. IOS recorded the supragingival preparations more precisely. Two further investigations have also demonstrated higher reproducibility for supragingival finishing lines [17,18]. Divergent literature states that the supra- and epigingival margins can be scanned without significant differences. In that mentioned in vitro study, supragingival finishing lines were made visible by gingival retraction [23]. The significant difference between the epi- and supragingival margins could be attributed to the absence of gingival retraction. Sufficient soft-tissue management is a crucial success factor and, therefore, should be ensured in clinical routine.

Based on the results of this in vitro investigation, it can be recommended that, to ensure a higher predictability of digital impression-taking in clinical routine, the finishing line must be clearly visible, with healthy gingiva surrounded a full 360°. Therefore, complete-crown finishing lines should be prepared supragingivally whenever possible using IOS [19,24], including proper soft-tissue management during impression taking, which remains a crucial success factor for any kind of impression technique [25] until future technology can provide novel IOS possibilities for scanning through tissue and liquids.

Today, IOS requires a direct line of sight on the object being scanned, and a minimal distance of 0.5 mm between adjacent teeth seems to be the critical threshold for the optical resolving power [17]. Otherwise, the IOS software takes over to calculate the preparation margins virtually, instead of capturing the intraoral situation with optical precision. Not all surfaces of a tooth seem to be recorded with the same accuracy; for example, distal and lingual surfaces have shown the lowest accuracy [26,27]. Finally, the complexity of the geometry to be scanned has an impact on the accuracy as well. Supragingival complete crown preparations have demonstrated significantly better results than intracoronal inlay preparations using different IOS systems [28]. For more complex preparations, e.g., for post copings or adhesive attachments, capturing with IOS is currently not feasible.

However, what are the limitations of digital impressions for treatment with complete crown restorations? Do any influencing factors affect the successful use of IOS in clinical routine?

Based on the results of the present trial, conventional crown preparation designs can be applied with a digital capturing by IOS while considering minimal invasiveness. It is possible to focus on the desired requirements for single-unit restorations in everyday clinical practice. The anatomical position and morphology of the area to be restored must be analyzed first. Moreover, the selected material has to be considered when selecting the preparation characteristics. Basically, the following parameters have been summarized for metal-based complete crowns: (i) convergence angle between two opposing prepared axial surfaces in the range of 10° to 22°; (ii) retentive vertical surfaces of at least 3 mm and a height-to-diameter ratio of at least 0.4 to provide adequate resistance form; (iii) teeth should be reduced uniformly to facilitate esthetic dental work, as well as anatomically to keep the teeth’s characteristic geometric shape and to avoid pulp trauma [19].

The translation from in vitro to in vivo always involves difficulties. The presented trial setting reflects ideal and constant conditions. The clinical real-world scenario has to tackle multi-factor challenges such as irregular tooth preparations in terms of design and distance to the gingival margin, different dental surfaces, perfused soft tissue, saliva and sulcus fluids, limited access in the oral cavity, and patient movement. It was also not possible to work with gingival retraction in this in vitro setting. This study was carried out in a stable single-jaw setting using a typodont model with ideally prepared artificial abutment teeth. The absence of saliva, tongue, mouth opening, and individual patient anatomy simplified IOS scanability [29]. The impact of mouth opening, in particular, needs be further investigated for mandibular impressions in vivo. During IOS, patients must maintain an extensive mouth opening for a longer time compared to the conventional approach. This could lead to slight deformations of the mandible [30,31]. With the conventional method, the mouth must only initially be opened wide; however, during the setting time of the material, the patient can almost rest in a relaxing position. In vivo, this could lead to deviations in scanning accuracy between anterior and posterior areas, which could not be detected in this in vitro setting. Additionally, only two IOS devices were used, which reduces the power of generalization, and the operators could not be blinded for the intervention and the type of scanner used. For further studies, it would be useful to include a greater variety of IOS scanners and also to perform subgingival preparations in vivo, where appropriate soft-tissue management could be applied.

## 5. Conclusions

Within the limitations of this study, the following can be concluded:(1)the overall accuracy for all abutment teeth was very high, without significant differences in the performance of 3Shape Trios 3 Pod versus Cerec Primescan AC;(2)the supragingival finishing lines were captured significantly better than the epigingivally located margins using IOS. If the clinical situation allows, a supragingival margin should be chosen accordingly; (3)the tooth morphology seems to be a negligible factor for IOS accuracy in terms of single-unit complete crown restorations.

## Figures and Tables

**Figure 1 jcm-09-03984-f001:**
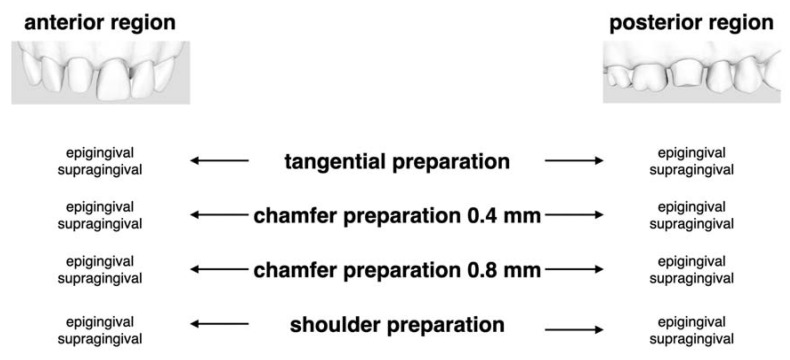
Trial setting: Four different preparation designs in anterior and posterior regions separated for epi- and supragingival finishing lines.

**Figure 2 jcm-09-03984-f002:**
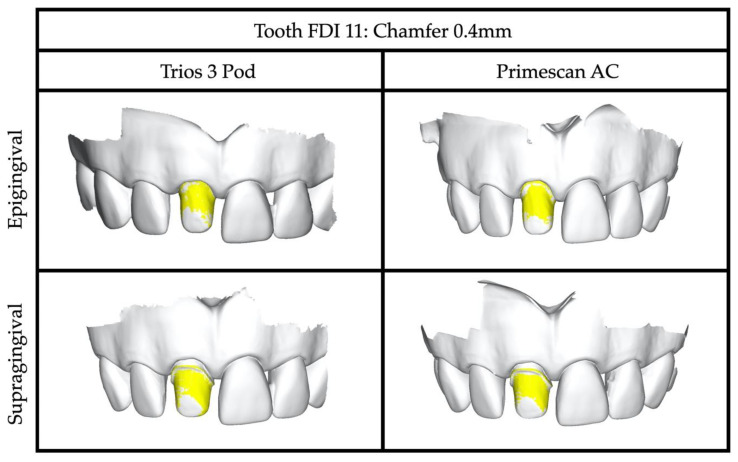
3D color mapping depicting sample abutment teeth in position FDI 11 with epi-/supragingival 0.4 mm chamfer captured with Trios 3/Primescan AC and superimposition to the corresponding references (Final Surface, GFaI e.V., Berlin, Germany).

**Table 1 jcm-09-03984-t001:** Anterior tooth morphology: Deviation (in μm) of IOS-STLs compared to the reference STLs summarizing mean values of 80% quantiles, including standard deviations (SD) of the different preparation designs separated for epi- and supragingival finishing lines (^a–f^
*p* < 0.05).

		Trios 3 Pod	Primescan AC
Epigingival	Tangential	34 ± 6	35 ± 5
Chamfer 0.4 mm	^a^ 38 ± 4	^b^ 40 ± 6
Chamfer 0.8 mm	^c^ 42 ± 5	^d^ 45 ± 6
Shoulder	^e^ 48 ± 5	^f^ 50 ± 5
Supragingival	Tangential	30 ± 1	31 ± 2
Chamfer 0.4 mm	^a^ 28 ± 3	^b^ 26 ± 2
Chamfer 0.8 mm	^b^ 29 ± 3	^d^ 30 ± 3
Shoulder	^e^ 40 ± 6	^f^ 39 ± 5
(^a^ *p* = 0.0036, ^b^ *p* = 0.001, ^c^ *p* = 0.0013, ^d^ *p* = 0.0008, ^e^ *p* = 0.0008, ^f^ *p* = 0.0025)

**Table 2 jcm-09-03984-t002:** Posterior tooth morphology: Deviation (in μm) of IOS-STLs compared to the reference STLs summarizing mean values of 80% quantiles, including standard deviations (SD) of the different preparation designs separated for epi- and supragingival finishing lines (^a–f^
*p* < 0.05).

		Trios 3 Pod	Primescan AC
Epigingival	Tangential	30 ± 4	31 ± 4
Chamfer 0.4 mm	^a^ 40 ± 6	^b^ 39 ± 4
Chamfer 0.8 mm	^c^ 39 ± 4	^d^ 41 ± 5
Shoulder	^e^ 34 ± 4	^f^ 36 ± 5
Supragingival	Tangential	29 ± 3	30 ± 3
Chamfer 0.4 mm	^a^ 28 ± 3	^b^ 32 ± 3
Chamfer 0.8 mm	^c^ 27 ± 2	^d^ 27 ± 1
Shoulder	^e^ 21 ± 2	^f^ 20 ± 2
(^a^ *p* = 0.0018, ^b^ *p* = 0.0128, ^c^ *p* = 0.0018, ^d^ *p* = 0.001, ^e^ *p* = 0.0013, ^f^ *p* = 0.0006)

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
