# Peer review of "Influence of Preparation Design, Marginal Gingiva Location, and Tooth Morphology on the Accuracy of Digital Impressions for Full-Crown Restorations: An In Vitro Investigation"

_jcm, 2020, doi:10.3390/jcm9123984_

Round 1

Reviewer 1 Report

The goal of the presented article was to analyze the intraoral optical scanning (IOS) accuracy for full-crown restorations in two different tooth morphologies (anterior vs posterior), design and location of the finish margins. Also two different IOS systems (Trios vs Primescan) were evaluated. 

I would like to thank the authors for this interesting work. The topic is of clinical relevance and the study design was overall appropriate with the limitations of an in-vitro methodology being recognized and described.

The following comments as listed below should be addressed by the authors: 1- Abstract Line 18: Change "full-arch" for "complete-arch" to suit better with the glossy of psothodontic terms   2- Introduction It is overall missing in the introduction relevant literature of clinical and in-vitro studies that have studied the topic before (influence of finish margin location and different IOS systems).   3- Material and methods

Who was the operator that manually finalized preparations with diamond burs and Sof-Lex discs? Was it the same one who performed the digital impressions and measurements analysis? My suggestion is to add the level of experience of the operator as well as the initials for each step performed by each operator.

When using the best-fit algorithm, what was the minimal tolerance accepted in the software? I suggest to add more details of this step since it is important to guarantee an accurate analysis of the real deviations encountered.

4- Results 

This sections is well presented and easy to follow.

5- Discussion

It is overall missing further comparison with clinical and in-viro studies that evaluated the influence of margin location and IOS systems. 

A clinical study by Koulivand et al. 2020 found that finish line positions had no significant effect on the fit of coping obtained after digital impressions. Was it due to effective gingival retraction. Please expand the discussion on this aspect and the limitation that retraction was not attempted.

Koulivand S, Ghodsi S, Siadat H, Alikhasi M. A clinical comparison of digital and conventional impression techniques regarding finish line locations and impression time. J Esthet Restor Dent. 2020;32:236–243.

A limitation that should be added is that operators could not be blinded to the intervention and type of scanner used.

6- Conclusion

In the results, authors reported that supragingival finishing lines demonstrated significantly higher accuracy than epigingival margins when comparing preparation design amongst each other (P < 0.05), while in the conclusion authors were more general "the clinical challenge for the treatment with complete crowns following digital impressions is the location of the finish line related to the gingival margin". Please elaborate regarding this matter.

Author Response

Reviewer: 1

Comments to the Author

The goal of the presented article was to analyze the intraoral optical scanning (IOS) accuracy for full-crown restorations in two different tooth morphologies (anterior vs posterior), design and location of the finish margins. Also two different IOS systems (Trios vs Primescan) were evaluated. 

I would like to thank the authors for this interesting work. The topic is of clinical relevance and the study design was overall appropriate with the limitations of an in-vitro methodology being recognized and described.

RESPONSE

Thank you very much for your time to review our manuscript and for your help in improving the quality of the manuscript.

The following comments as listed below should be addressed by the authors: 

1- Abstract 

Line 18: Change "full-arch" for "complete-arch" to suit better with the glossy of prosthodontic terms

RESPONSE

Thank you very much. We changed the term accordingly (Text change: P. 1; L. 18).

2- Introduction

It is overall missing in the introduction relevant literature of clinical and in-vitro studies that have studied the topic before (influence of finish margin location and different IOS systems).

RESPONSE

The scientific literature was screened and supplemental references were inserted at the appropriate text passages (Text change: P. 2; L. 57-59).

3- Material and methods

Who was the operator that manually finalized preparations with diamond burs and Sof-Lex discs? Was it the same one who performed the digital impressions and measurements analysis? My suggestion is to add the level of experience of the operator as well as the initials for each step performed by each operator.

RESPONSE

All practical work steps, such as finalization of preparations with diamond burs and sof-lex discs as well as intraoral scanning, were done by the same clinical operator (S.B.), a postgraduate prosthodontic resident and step by step supervised by a senior clinician and board-certified prosthodontist (J.M.). (Text changes: P. 2; L 81-83 and P. 3, L. 106-107).

When using the best-fit algorithm, what was the minimal tolerance accepted in the software? I suggest to add more details of this step since it is important to guarantee an accurate analysis of the real deviations encountered.

RESPONSE

We added more details related to the best-fit algorithm for accuracy analysis (Text changes: P. 3; L. 110-119).

4- Results 

This section is well presented and easy to follow.

RESPONSE

Thank you very much.

5- Discussion

It is overall missing further comparison with clinical and in-vitro studies that evaluated the influence of margin location and IOS systems. 

A clinical study by Koulivand et al. 2020 found that finish line positions had no significant effect on the fit of coping obtained after digital impressions. Was it due to effective gingival retraction? Please expand the discussion on this aspect and the limitation that retraction was not attempted.

Koulivand S, Ghodsi S, Siadat H, Alikhasi M. A clinical comparison of digital and conventional impression techniques regarding finish line locations and impression time. J Esthet Restor Dent. 2020;32:236–243.

A limitation that should be added is that operators could not be blinded to the intervention and type of scanner used.

RESPONSE

Thank you very much. We integrated the study mentioned above and extended the discussion accordingly. The mentioned limitation is justified and has been added. (Text change: P. 5; L. 161-163 / P. 6 L. 200 and 211).

6- Conclusion

In the results, authors reported that supragingival finishing lines demonstrated significantly higher accuracy than epigingival margins when comparing preparation design amongst each other (P < 0.05), while in the conclusion authors were more general "the clinical challenge for the treatment with complete crowns following digital impressions is the location of the finish line related to the gingival margin". Please elaborate regarding this matter.

RESPONSE

Thank you for your advice. The second conclusion has been reworded accordingly. (Text change: P. 6; L. 218-220).

Reviewer 2 Report

Dear authors,

This is a well-written manuscript of current interest. There are some issues that could be addressed to improve the manuscript.

Introduction:

Line 49: You state that the finish line design for tooth preparation have remained a crucial aspect for the abutment tooth. What do you base this statement on? Reference?

Discussion:

Lines 137-152. In my opinion, this is more of an introduction rather than a discussion offering explanations for you findings. The same applies largely to lines 157-162.

You do not use any form of retraction cord etc. as this is an in-vitro study. Do you think that the problem remains in the clinical situation where this would be applied? Please mention/discuss.

Limitations are well discussed.

Conclusion:

The second conclusion “the clinical challenge for the treatment with complete crowns following digital impressions is the location of the finish line related to the gingival margin “ must be adjusted as you have not performed a clinical trial and thus can only hypothesize about possible clinical importance, and there are few other studies supporting the statement.

Why not instead be more specific regarding your findings on finish lines “Supragingival finishing lines demonstrated significantly higher accuracy”?

References:

The authors cite themselves no less than 6 times. The topic of these references fit the respective texts and are appropriate, but there are other groups that could be mentioned.

Author Response

Reviewer: 2

Comments to the Author

Dear authors,

This is a well-written manuscript of current interest. There are some issues that could be addressed to improve the manuscript.

RESPONSE

Thank you very much for your time to review our manuscript and for your help in improving the quality of the manuscript.

Introduction:

Line 49: You state that the finish line design for tooth preparation have remained a crucial aspect for the abutment tooth. What do you base this statement on? Reference?

RESPONSE

References were inserted at the appropriate place (Text change: P. 1; L. 61).

Discussion:

Lines 137-152. In my opinion, this is more of an introduction rather than a discussion offering explanations for you findings. The same applies largely to lines 157-162.

You do not use any form of retraction cord etc. as this is an in-vitro study. Do you think that the problem remains in the clinical situation where this would be applied? Please mention/discuss.

Limitations are well discussed.

RESPONSE

The corresponding lines 137-152 were partially deleted or transferred to the Introduction. The discussion was extended to include the aforementioned content related to “gingival retraction”. (Text changes: P. 1; L. 46-55. and P. 5 L. 161-165).

Conclusion:

The second conclusion “the clinical challenge for the treatment with complete crowns following digital impressions is the location of the finish line related to the gingival margin“ must be adjusted as you have not performed a clinical trial and thus can only hypothesize about possible clinical importance, and there are few other studies supporting the statement.

Why not instead be more specific regarding your findings on finish lines “Supragingival finishing lines demonstrated significantly higher accuracy”?

RESPONSE

Thank you for your recommendation. The second conclusion has been reworded accordingly. (Text change: P. 6; L. 218-220).

References:

The authors cite themselves no less than 6 times. The topic of these references fit the respective texts and are appropriate, but there are other groups that could be mentioned.

RESPONSE

Additional references were inserted at the appropriate place (Text changes: throughout the entire manuscript and the reference list).